# Numerical Simulation and Experiment Study on the Characteristics of Non-Darcian Flow and Rheological Consolidation of Saturated Clay

**Zhongyu Liu [1,\*], Yangyang Xia [1,2]** **, Mingsheng Shi [2,\*], Jiachao Zhang [1] and Xinmu Zhu [1]**

[1]  School of Civil Engineering, Zhengzhou University, Zhengzhou 450001, China
[2]  School of Water Conservancy and Environment, Zhengzhou University, Zhengzhou 450001, China
\*  Correspondence: zhyliu@zzu.cn (Z.L.); sms315@126.com (M.S.)

**Abstract:** To investigate the characteristics of the non-Darcian water flow through saturated clay and one-dimensional rheological consolidation behaviors of the soil in the Henan Province, we conducted constant-head permeability and one-dimensional rheological consolidation tests with one-way drainage using improved permeameter and oedometer tests, respectively. We then used Hansbo's flow equation to classify the permeability test results and one-dimensional rheological consolidation equation combined with unified hardening (UH) constitutive model considering time effect was introduced to simulate the oedometer test results. The obtained results showed that the improved constant-head permeability test device was suitable for saturated clays, and that the UH constitutive model and Hansbo's flow equation had good applications for the saturated clays investigated in this experiment.

**Keywords:** soil mechanics; saturated clay; non-Darcian flow; rheological consolidation; permeability

---

## 1. Introduction

Terzaghi's [1] one-dimensional consolidation theory has been widely employed by researchers since its introduction. Especially in areas with saturated clay, researchers have found that settlement values predicted by Terzaghi's theory have large deviations from measured values, mainly because this theory ignores the influence of non-Darcian flow of pore water seepage and the rheological properties of saturated clays. Therefore, many researchers have conducted studies on saturated clays considering the above two factors. As early as 1898, King [2] found a deviation from Darcy flow in saturated clays, which was unfortunately unrecognized by the academic community at the time. In 1960, Hansbo [3] also witnessed this phenomenon, which he called the non-Darcian flow, and proposed a corresponding mathematical model using experimental results. The non-Darcian flow phenomenon in saturated clays was subsequently confirmed by many researchers around the world, and different mathematical equations were proposed for it. For example, Swartzendruber [4] proposed the non-Newtonian index flow model, Slepicka [5] proposed the exponential flow model, Miller et al. [6] proposed the threshold gradient flow model, and Qi et al. [7] proposed the broken line flow model. At the same time, Dubin et al. [8], Wang et al. [9], Deng et al. [10], Sun et al. [11], performed experiments showing that seepage in the relevant clays deviated from Darcy's law in varying degrees. Wang et al. [9] attributed the deviation to the combined action of gravity water, capillary water, and weak bound water in saturated clay under different hydraulic gradients. Currently, Hansbo's [3] flow theory is most commonly used, and has been applied to one-dimensional [12–15] and two-dimensional [16] consolidations, as well as sand drains [17], further confirming the necessity of considering non-Darcian flow in the analysis of saturated clay consolidation.

All these previous studies related to clay consolidation mostly focused on non-linearity [18,19], anisotropy [20], large-strain [21], time-dependent loading [22], and so on, while relatively few studies are related to the rheological properties of clay. Early studies [23–25] believe that the rheological properties of clay are caused by the viscosity of water around soil particles, the plastic adjustment of soil particles' structure, the retention and creep due to shear stress, and hydrostatic pressure. However, there was a disagreement as to whether rheology should be taken into account during consolidation. Buisman [23] believed that rheological or secondary consolidation occurred after completing the main consolidation, while Tayor et al. [24] suggested that secondary consolidation occurred throughout the consolidation process. Bjerrum [26] confirmed Tayor's theory through experiments and described the creep deformation of soil using a simple element model. Then, with continuous improvement in research methods, researchers around the world carried out in-depth research on the rheological properties of saturated clay based on one-dimensional rheological consolidation tests considering the effects of external load, loading ratio, preloading, drainage conditions, etc. First, Bjerrum [27] believed that the secondary consolidation coefficient was linearly related to the external load. However, Nash et al. [28] and Chen et al. [29] showed that the secondary consolidation coefficient first increased and then decreased with the increase in external load. Yin et al. [30] and Reddy et al. [31] further confirmed the above conclusion. Unlike Nash et al. [28], Chen et al. [29], Yin et al. [30], and Reddy et al. [31], it was found that the peak value of the secondary consolidation coefficient did not necessarily appear near pre-consolidation pressure, which Yin et al. [30] attributed to the effect of loading ratio and preloading. More in depth research ultimately demonstrated a high degree of nonlinearity in the deformation of saturated clay. Yin [32,33], Yao [34,35], and others have proposed some elastic visco-plastic (EVP) models. The unified hardening (UH) constitutive model was applied in one-dimensional rheological consolidation analysis, and successfully explained the increase in excess pore pressure near impervious surfaces at the initial stage of rheological consolidation. Compared to the EVP model proposed by Yin [32], the UH model proposed by Yao [34] has many advantages, such as fewer model parameters and a clearer physical concept, and thus could be simply determined by routine experiments.

In summary, in order to further explore the permeability and rheological consolidation characteristics of saturated clay, traditional permeability and one-dimensional consolidation test devices, as well as saturated remolded clay were improved based on previous works. A series of permeability and rheological consolidation tests were carried out, and their results were compared with fitting results obtained from Hansbo's flow model and theoretical prediction results from the UH rheological consolidation model in order to verify the applicability of both models to saturated clay.

## 2. Experimental Study on the Permeability Characteristics of Saturated Clay

### 2.1. Constant-Head Permeability Test Device

Currently, there are two main types of permeability test methods, namely constant-head and falling-head [36]. Previous studies on saturated clay permeability employ the falling-head tests, but as shown in Figure 1, because of the compactness and low permeability of clay, the falling-head permeability test requires a long time, and the errors caused by human and environmental factors cannot be avoided during the test. However, it is noteworthy that the constant-head permeability test requires a very short time, but this test is primarily used to measure sand or granular soil with better permeabilities, and mostly to determine water volume using measuring cylinder or beaker. For clays with insignificant amounts of water flow and low permeability, it is impossible to measure water volume cannot be accurately measured in a short time; therefore, we improved the conventional constant-head permeability test device.

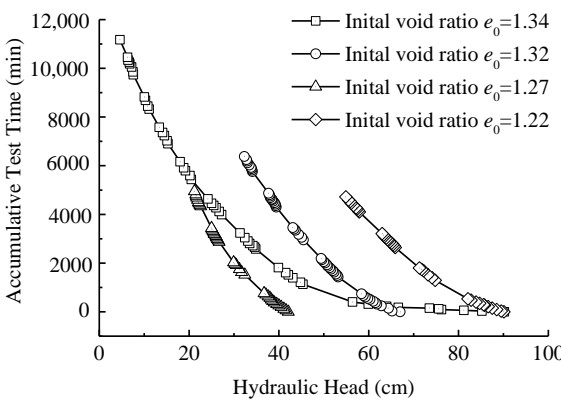

**Figure 1.** Falling-head permeability test results.

Figure 2 shows a schematic diagram of the improved constant-head test device proposed in this paper. Its main difference from the conventional constant-head permeability device is that in the modified device, a graduated U-tube with a 4 mm inner diameter was equipped with a video device to achieve accurate measurement of insignificant amounts of water flow in a short period of time. On the other hand, test head measurement was realized by combining the constant-head control device and calibration plate, which reduces test time and test error, and greatly improves test efficiency.

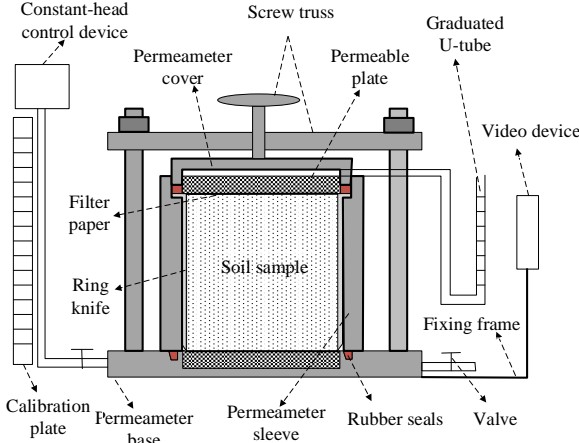

**Figure 2.** Schematic diagram of the improved constant-head permeability test device.

### 2.2. Permeability Test and Data Processing

The clay sample used in this paper was collected from a certain place in the Henan Province, at a 2.5 m depth. The basic physical parameters of the soil are summarized in Table 1. Since natural undisturbed soil contains more impurities and it was not uniform, saturated remolded clay was used in this paper.

**Table 1.** Physical properties of soils.

| Specific Gravity, $G_s$ | Liquid Limit, $w_L$ (%) | Plastic Limit, $w_P$ (%) | Plastic Index, $I_P$ (%) |
|:---:|:---:|:---:|:---:|
| 2.71 | 39.4 | 22.3 | 17.1 |

First, six permeability samples with different void ratios were prepared using a self-made saturated remolded clay sampler [37], according to the Standard for Soil Test Method (GB/T50123-1999) [36]. The height and diameter of the sample were 4 and 6.18 cm, respectively, and they were labeled SL-1 to SL-6. After vacuum saturation, permeability tests were carried out using the device described

in Figure 2. The details of the test procedures were as follows: (1) Coat a layer of Vaseline on the inner wall of the permeameter sleeve, and attach a filter paper to the upper and lower surfaces of the saturated sample with cutting ring (to avoid being covered by Vaseline when the sample was installed, which could affect the permeability of the sample). Then, the sample was loaded into the permeameter sleeve, the excess Vaseline was wiped out and upper and lower filter papers were removed. Finally, permeameter sleeve was installed on the base and tighten the screw was tightened. (2) Constant head control device was adjusted and excess air inside the permeameter was discharged. Then, liquid levels were adjusted to calibration lines in the graduated U-tubes. (3) The left valve on the bottom of the permeameter was opened and the video device on the right side was turned on. it was left until liquid level in the U-tube reached the next calibration line and the test was stopped(where one scale was 0.034 mL, therefore the test time was shorter). Then, the constant head control device was adjusted using the same method and the corresponding seepage velocities under different constant pressure heads were recorded. Finally, *v–i* curves were obtained by connecting these experimental points.

Using the nonlinear fitting toolbox in Origin software and Hansbo's [3] flow mathematical equations shown in Equation (1), permeability test data were segmentally fitted as shown in Figure 3. Test results were compared with those obtained from Hansbo's flow fitting. The basic parameters of soil sample and Hansbo's flow are summarized in Table 2.

$$v = \begin{cases} ci^m, & i < i_1 \\ k(i - i_0), & i \geq i_1 \end{cases} \tag{1}$$

where $i_0 = i_1(m-1)/m$, $c = k/(mi_1{}^{m-1})$; $c$ and $k$ are the coefficient of permeability of the exponential flow at low gradients and the linear flow at high gradients, respectively; $m$ is the exponent of exponential flow at low gradients; $i_0$ is the initial the hydraulic gradient of the linear percolation section; and $i_1$ is the threshold hydraulic gradient for the linear relationship. When $i_0 = 0$ or $m = 1$, Equation (1) degenerates into Darcy's flow law.

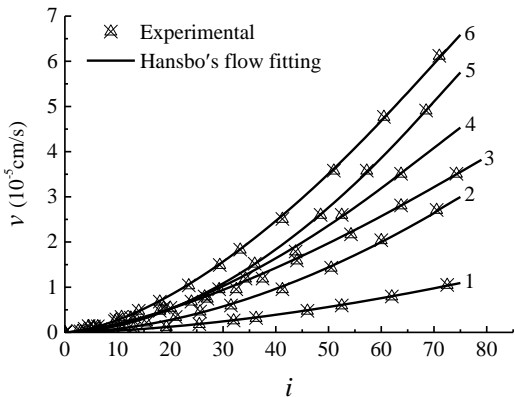

**Figure 3.** Comparison of experimental results and those obtained from Hansbo's flow fitting.

**Table 2.** Hansbo's flow fitting parameters.

| Soil Sample Number | $e_0$ | $k$ (cm/s) | $i_0$ | $m$ | $R^2$ |
|---|---|---|---|---|---|
| SL-1 | 1.26 | $6.73 \times 10^{-8}$ | 22.20 | 1.45 | 0.985 |
| SL-2 | 1.29 | $1.28 \times 10^{-7}$ | 23.54 | 1.58 | 0.978 |
| SL-3 | 1.36 | $2.31 \times 10^{-7}$ | 27.81 | 1.63 | 0.964 |
| SL-4 | 1.44 | $6.85 \times 10^{-7}$ | 31.25 | 1.81 | 0.972 |
| SL-5 | 1.47 | $8.95 \times 10^{-7}$ | 24.37 | 1.64 | 0.967 |
| SL-6 | 1.50 | $1.29 \times 10^{-6}$ | 30.41 | 1.83 | 0.956 |

where $e_0$ is the initial void ratio , $k$ is the permeability coefficient of soil samples, $R^2$ is the correlation coefficient.

The developed fitting method was as follows: (1) If there were $n$ experimental points, first Origin software was used for non-linearly fitting of 0 to g (where $0 < g < n$) experimental points based on $v = ci^m$ to obtain the values of $c$ and $m$. (2) Equation $v = cmi_1^{m-1}[i - i_1(m-1)/m]$ was obtained by substituting $c$ and $m$ into formula $v = k(i - i_0)$, then the g to $n$ experimental points were non-linearly fitted through equation $v = cmi_1^{m-1}[i - i_1(m-1)/m]$ by Origin software, and the value of $i_1$ was obtained. (3) By substituting the values of $c$, $m$, and $i_1$ into the Equation (1), the calculation points of Hansbo's flow mathematical model were obtained, and the correlation between experimental and the calculation values was compared to obtain maximum $R^2$. Otherwise, the above methods were repeated at 0 to g + 1 and g + 1 to $n$ experimental points. where $e_0$ is the initial void ratio, $k$ is the permeability coefficient of soil samples, $R^2$ is the correlation coefficient.

Figure 3 and Table 3 show that the results obtained from Hansbo's flow equation fitting and test results had a high degree of fitting with a correlation coefficient $R^2$ above 0.95. Hansbo's flow parameter $m$ was also within the ranges suggested by Hansbo [3], Dubin et al. [8], and Sun et al. [11] (between 1 and 2). However, due to the lack of relevant experimental data, there is no unified statement on the range of $i_0$ in current studies, Hansbo [3] stated that $i_0$ ranged from 1 to 4; Dubin et al. [8] reported it was between 3 and 25, and Sun et al. [11] reported it to be between 0.5 and 14, and the results obtained in this paper were basically within the above range, which proves the applicability of Hansbo's flow model to clay samples.

**Table 3.** Basic properties of the rheological consolidation test sample.

| Sample Number | Specific Gravity, $G_s$ | Bulk Density, $\rho$ (g/cm$^3$) | Water Content, $w$ (%) | Void Ratio, $e_0$ |
|---|---|---|---|---|
| LB-1 | 2.71 | 1.738 | 47.77 | 1.30 |
| LB-2 | 2.71 | 1.864 | 34.14 | 0.95 |

Taylor [38] proposed a linear relationship between the logarithm of $k$ and void ratio $e$ as shown in Equation (2):

$$\log k = \log k_0 - (e_0 - e)/C_k \tag{2}$$

where $k_0$ is the initial permeability coefficient corresponding to a initial void ratio $e_0$ and $C_k$ is the permeability index.

In order to prove the applicability of Equation (2) to the clay samples studied in this paper, void ratio $e$ and permeability coefficient $k$ listed in Table 2 were nonlinearly fitted according to Equation (2), and the obtained results are shown in Figure 4. The experimental data fit values satisfied the $e$-log$k$ relationship, and the permeability index $C_k = 0.193$ was also in the reasonable range of permeation index given by Berry [39] ($C_k$ value between 0.02 and 5.00). Therefore, it was confirmed that Equation (2) is applicable to the clay samples studied in this paper, i.e., log$k$ and the void ratio $e$ were linearly related.

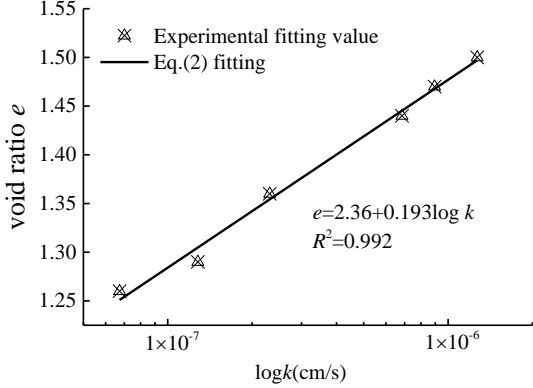

**Figure 4.** The relationship between the permeability coefficient $k$ and void ratio $e$.

## 3. Experimental Study on Rheological Consolidation Characteristics of Saturated Clay

### 3.1. One-Dimensional Rheological Consolidation Test

In order to improve test accuracy, the TWJ-1 data collector device was introduced based on Sun's [11] permeability-consolidation cross-test device. Displacement and pore water pressure at the bottom of the sample were measured by connecting the device to a computer. Figure 5 shows the schematic diagram of the improved consolidation device.

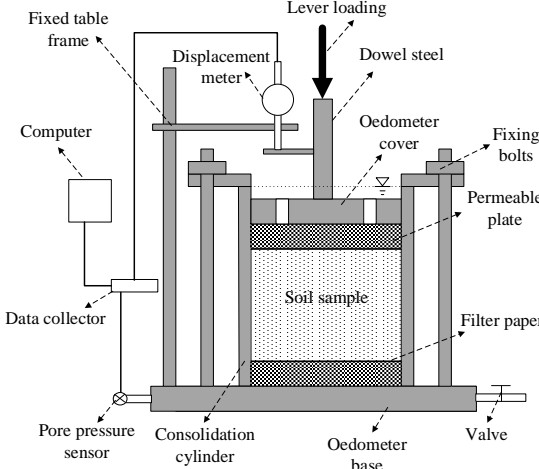

**Figure 5.** Schematic diagram of the improved oedometer.

First, two consolidated samples were prepared as described in Section 2.2 with sample height and diameter of 2 and 6.18 cm, respectively. The basic physical parameters of the sample are summarized in Table 2. In order to simulate the rheological consolidation process of a natural saturated clay layer, the one-dimensional rheological consolidation test with upward drainage was used in this work. The loading and unloading scheme shown in Table 3 were applied based on Yin et al. [32] and Feng et al. [33]. Detailed test procedures were as follows: (1) the prepared saturated clay samples LB-1 and LB-2 were loaded into the consolidation cylinder, respectively, as shown in Figure 5. Filter paper and permeable plate are placed on the top and bottom of the samples. (2) Displacement meter and pore pressure sensor were adjusted using the data collection sofware. Then, loading was started when the displacement sensor gave values lower than 0.1 mm in a day or the pore pressure dissipated more than 95%, the next stage of loading or unloading was carried out according to Table 4. (3) According to the displacement sensor reading, the vertical settlement deformation *S* of the soil sample was calculated, and then the corresponding void ratio *e* was obtained according to Equation (3).

$$\log k = \log k_0 - (e_0 - e)/C_k \tag{3}$$

where $e_0$ is the initial void ratio corresponding to the initial thickness $H_0$ of the soil sample.

**Table 4.** Loading procedures in oedometer tests.

| Order | Loading (kPa) |
|---|---|
| First loading | 0-25-50-100-200 |
| First Unloading | 200-100-50 |
| Second Loading | 50-100-200-400 |
| Second Unloading | 400-200-100-50 |
| Third Loading | 50-100-200-400-800 |
| Third Unloading | 800-400-200-100-50 |
| Fourth Loading | 50-100-200-400-800-1600 |

*3.2. One-Dimensional Rheological Consolidation Test Data Processing*

　　Figure 6 shows the relationship between void ratio and time-dependent loading under first-stage loading conditions. As can be seen in the figure, the obtained results were consistent with those obtained by Yin et al. [32] and Feng et al. [33] for Hong Kong clay samples; the *e*-log*t* curve had an inverse "*S*" shape and its tail section was elongated with the increase of vertical load. At the same time, the *e*-log*t* curve depended to some extent on the initial void ratio of the sample. As can be seen from Figure 6, by the decrease of initial void ratio, the tail section of *e*-log*t* curve became steeper and longer times were required for deformation stabilization. In addition, the slope of the tail section of the *e*-log*t* curve reflected the secondary consolidation coefficient. Table 5 summarizes the secondary consolidation coefficient values obtained by fitting the tail section of the *e*-log*t* curve under different loads. These results are similar to those obtained by Nash et al. [28] and Chen et al. [29], where the secondary consolidation coefficient was first increased and then decreased with the increase of external load. Smaller initial void ratios of the sample gave greater consolidation coefficients. This could be attributed to the stability of clay structure at the particle's contact surface [40].

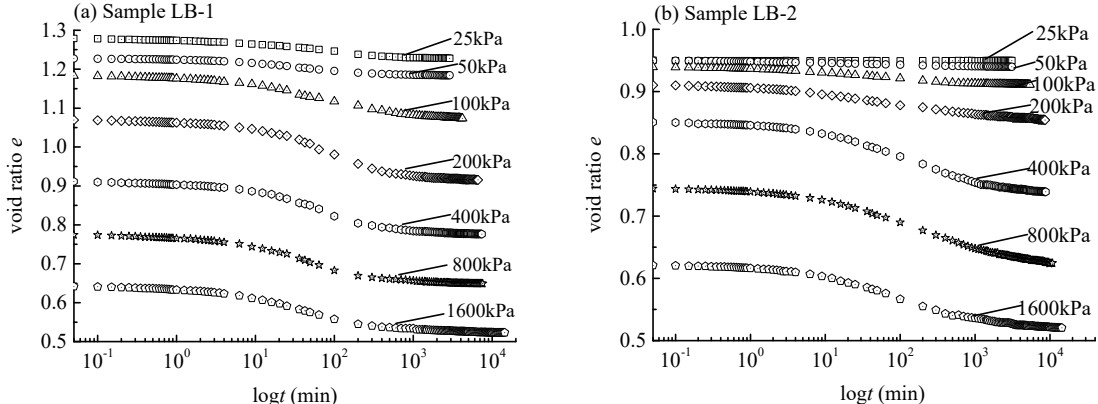

**Figure 6.** *e*-log*t* curves of samples LB-1 and LB-2 under primary loading condition.

**Table 5.** Fitting results of secondary consolidation coefficient $C_\alpha$ under primary loading conditions.

| Soil Sample Number | Loading (kPa) | | | | | | |
| --- | --- | --- | --- | --- | --- | --- | --- |
| | 25 | 50 | 100 | 200 | 400 | 800 | 1600 |
| LB-1 | 0.0040 | 0.0050 | 0.0120 | 0.0110 | 0.0091 | 0.0085 | 0.0080 |
| LB-2 | 0.0009 | 0.0015 | 0.0030 | 0.0092 | 0.0135 | 0.0208 | 0.0134 |

　　Figure 7 shows the *e*-log *p* curves of samples LB-1 and LB-2. The curve was fitted according to the Standard for Soil Test Method (GB/T50123-1999) to obtain the values of $C_c$ and $C_s$, and it was found that the average value of $C_c$ for the two samples was 0.42. The values of $C_c/C_s$ for samples LB-1 and LB-2 were 4.32–4.79 and 5.20–7.65, respectively. The corresponding $C_c/C_s$ value of saturated clay was between 4.32 and 7.65, which was also obtained by Li [41] based on the GDS consolidation instrument who found $C_c/C_s$ value of 3.65 to 7.88 for clay samples collected from the Xiaoshan in Zhejiang province. Based on the above analysis, the effectiveness of the improved oedometer proposed in this paper was verified.

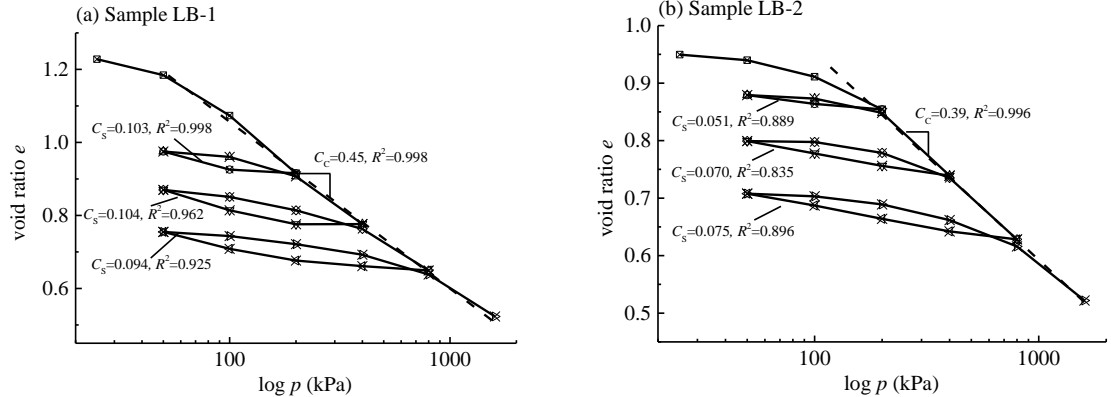

**Figure 7.** *e*-log *p* (vertical load) curves for samples LB-1 and LB-2.

## 4. Derivation of Governing Equations

### 4.1. UH Model Considering Time Effect

The above tests show that saturated soft clays have obvious rheological properties. In order to use this property in predicting consolidation behavior, a unified hardening (UH) model considering time effect was introduced [34,35] to describe the stress–strain relationship of soil. These relationships are expressed as:

$$d\varepsilon_v = \begin{cases} d\varepsilon_v^e + d\varepsilon_v^{sp} + d\varepsilon_v^{tp}, & d\sigma' \geq 0 \\ d\varepsilon_v^e + d\varepsilon_v^{tp}, & d\sigma' < 0 \end{cases} \tag{4}$$

where $d\varepsilon_v$ is the vertical strain increment and $d\varepsilon_v^e$, $d\varepsilon_v^{sp}$ and $d\varepsilon_v^{tp}$ are elastic, plastic and time-dependent viscous strain increments caused by stress, respectively, and can be expressed as:

$$d\varepsilon_v^e = C_s d\sigma' / [\ln 10(1 + e_0)\sigma'] \tag{5}$$

$$d\varepsilon_v^{sp} = \frac{C_c - C_s}{\ln 10(1 + e_0)} \frac{M^4}{M_f^4} \frac{d\sigma'}{\sigma'} \tag{6}$$

$$d\varepsilon_v^{tp} = C_\alpha dt / [\ln 10(1 + e_0)(t_\alpha + t_0)] \tag{7}$$

where $C_c$ is the compressed index, $C_s$ is the swelling index, $C_\alpha$ is the secondary consolidation coefficient, $t_\alpha$ is the aging time, $t_0$ is the unit time which can be 1 s, 1 min, 1 h, etc. According to actual situations, $M$ is the stress ratio at critical state and $M_f$ is the potential failure stress ratio, which can be expressed as:

$$M = 6\sin\varphi / (3 - \sin\varphi) \tag{8}$$

$$M_f = 6\left[\sqrt{\chi/R(1 + \chi/R)} - \chi/R\right] \tag{9}$$

$$(t_a + t_0)/t_0 = R^{-\alpha} \tag{10}$$

$$R = \frac{\sigma'}{p'_c} \exp\left(-\frac{\ln 10 \cdot (1 + e_0)\varepsilon_v^p}{C_c - C_s}\right) \tag{11}$$

where $\varphi$ is the internal friction angle of soil, $\chi = M^2/12(3 - M)$, $\alpha = (C_c - C_s)/C_\alpha$, and $R$ is a parameter reflecting the degree of over-consolidation. When time effect is not considered, the initial value $R_0$ is reciprocal of the over-consolidation ratio (OCR). $\varepsilon_v^p$ is the plastic strain produced by stress and time, $p'_c$ is the vertical stress at the intersection of rebound and instantaneous compression lines, similar to pre-consolidation pressure.

### 4.2. One-Dimensional Rheological Consolidation Equation of UH Model

First, the principle of effective stress is obtained as:

$$\sigma' = \sigma'_0 + p_0 - u \tag{12}$$

where $\sigma'_0$ is the initial effective stress, $p_0$ is the external loads, and $u$ is the excess pore water pressure. For both sides of Equation (12), partial derivatives could be obtained at the same time as:

$$\frac{\partial \sigma'}{\partial z} = -\frac{\partial u}{\partial z} \tag{13}$$

Based on continuous conditions of permeability we had:

$$\frac{\partial q}{\partial z} = -\frac{\partial \varepsilon_v}{\partial t} \tag{14}$$

where $q$ is the change of water quantity within the unit cell during an infinitesimal time period of $dt$ can be expressed as $q = vA$, $A$ is the area of water flowing through the unit cell.

Substituting Equations (1), (2), (4), and (13) into Equation (14) gives:

$$\frac{\partial}{\partial z}\left(\psi \frac{\partial \sigma'}{\partial z}\right) = \beta \frac{\partial \sigma'}{\partial t} + \eta \tag{15}$$

where $\eta = \gamma_w C_\alpha / [\ln 10 k_0 (1 + e_0)(t_\alpha + t_0)]$, $\psi = \begin{cases} \frac{\exp[-\ln 10 \varepsilon_v (1+e_0)/C_k]}{m(\gamma_w i_1)^{m-1}} \cdot \left|\frac{\partial \sigma'}{\partial z}\right|^{m-1}, & |i| \le i_1 \\ \exp[-\ln 10 \varepsilon_v (1 + e_0)/C_k], & |i| > i_1 \end{cases}$,

$\beta = \begin{cases} \frac{[C_S + (C_C - C_S)M^4/M_f^4]\gamma_w}{(1+e_0)k_0 \sigma' \ln 10}, & d\sigma' \ge 0 \\ \frac{C_S \gamma_w}{(1+e_0)k_0 \sigma' \ln 10}, & d\sigma' < 0 \end{cases}$

However, the initial and boundary conditions of the device shown in Figure 4 are expressed as:

$$\sigma'(z,0) = \sigma'_0, \quad 0 \le z \le H \text{ (initial condition)} \tag{16}$$

$$\sigma'(0,t) = p_0 + \sigma'_0, \quad t > 0 \text{ (permeable boundary)} \tag{17}$$

$$\left.\frac{\partial \sigma'}{\partial z}\right|_{z=H} = 0, \quad t > 0 \text{ (imperious boundary)} \tag{18}$$

### 4.3. Discretization of the Governing Equation

Due to the complexity of Equation (15), it was very difficult to obtain its analytical solution. Therefore, the numerical solutions can be obtained using the finite difference method (FDM), finite volume method (FVM), or finite element method (FEM). It was found that among these methods, the FVM was a numerical solution based on FDM, and absorbs the advantages of the FEM. Some researchers have studied this method [15,42] and their results showed that FVM as more advantages than FDM and FVM in terms of convergence and accuracy. Therefore, FVM was used in this paper to solve the problem.

To obtain numerical solutions for Equation (15) using FVM, as shown in Figure 8, the spatial domain $0 \le z \le H$ was firstly divided into $n$ thin layers with equal length $\Delta z$. Meanwhile, the time axis is divided by a series of time increments $\Delta t$. Where $b$th is the number of spatial nodes, $j$th is time discretized point. Then, within any time interval $\Delta t$ ($t_j$ to $t_{j+1}$), Equation (15) was integrated in the control volume of $b$th:

$$\int_{t_j}^{t_{j+1}} \int_{\Delta z} \frac{\partial}{\partial z}\left(\psi \frac{\partial \sigma'}{\partial z}\right) dz dt = \int_{t_j}^{t_{j+1}} \int_{\Delta z} \left(\beta \frac{\partial \sigma'}{\partial t} + \eta\right) dz dt \tag{19}$$

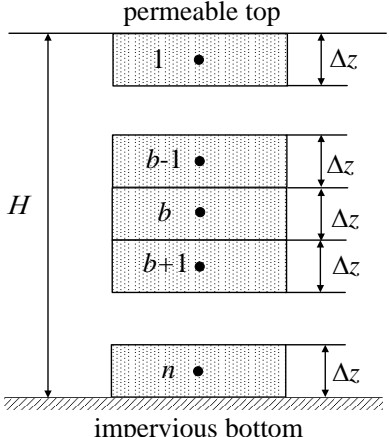

**Figure 8.** Discretized schematic of the soil layer.

Equation (19) can be rewritten as:

$$\int_{t_j}^{t_{j+1}} \left( \psi \frac{\partial \sigma'}{\partial z} \bigg|_B - \psi \frac{\partial \sigma'}{\partial z} \bigg|_D \right) dzdt = \int_{t_j}^{t_{j+1}} \int_{\Delta z} \left( \beta \frac{\partial \sigma'}{\partial t} + \eta \right) dzdt \tag{20}$$

where the subscripts $B$ and $D$ represent the lower and upper boundary points of the control volume, respectively.

For Equation (20), the first-order forward difference is used to approximate the right side of the equal sign $\partial \sigma' / \partial t$ and the central difference is used to approximate the left side of the equal sign $\partial \sigma' / \partial z$. To describe the relationship between $\sigma'_{b-1}$, $\sigma'_b$, $\sigma'_{b+1}$, and time, the effective stresses at $t_j$ and $t_{j+1}$ are usually weighted as a combination and then taken as the average value of the time from $t_j$ to $t_{j+1}$. Then, the time integral term on the left side of Equation (20) is calculated. Where the weight of $t_j$ is 0 and that of $t_{j+1}$ is 1. That is to say, Equation (20) can be written as:

$$\left[ \psi_{b+1/2}^{j+1} \left( \frac{\sigma'^{j+1}_{b+1} - \sigma'^{j+1}_b}{\Delta z} \right) - \psi_{b-1/2}^{j+1} \left( \frac{\sigma'^{j+1}_b - \sigma'^{j+1}_{b-1}}{\Delta z} \right) \right] \Delta t = \beta_b^{j+1}(\sigma'^{j+1}_b - \sigma'^j_b)\Delta z + \eta_b^{j+1}\Delta z \Delta t \tag{21}$$

Equation (21) is reorganized as:

$$\sigma'^{j+1}_b = \sigma'^j_b + \left[ \psi_{b+1/2}^{j+1}(\sigma'^{j+1}_{b+1} - \sigma'^{j+1}_b) - \psi_{b-1/2}^{j+1}(\sigma'^{j+1}_b - \sigma'^{j+1}_{b-1}) \right] \Delta t / (\beta_b^{j+1}\Delta z^2) - \eta_b^{j+1}\Delta t / \beta_b^{j+1} \tag{22}$$

where $b$ is the number of spatial nodes, $j$ is time discretized point, $x^j_b$ represents the value of $x$ variable at $z = (b-1/2)\Delta z$ and $t = j\Delta t$, $x^j_{b\pm 1/2}$ represents the value of $x$ variable at $z = (b-1/2 \pm 1/2)$ and $t = j\Delta t$, the same below.

With the above-mentioned method (FVM), Equations (16)–(18) can be discretized as:

$$\sigma'^0_b = \sigma'_0, \quad b = 1, 2, 3, \ldots, n \tag{23}$$

$$\sigma'^{j+1}_1 = \sigma'^j_1 + \left[ \psi_{3/2}^{j+1}(\sigma'^{j+1}_2 - \sigma'^{j+1}_1) - 2\psi_{1/2}^{j+1}(\sigma'^{j+1}_1 - p_0 - \sigma'_0) \right] \Delta t / (\beta_1^{j+1} \cdot \Delta z^2) - \Delta t \cdot \eta_1^{j+1} / \beta_1^{j+1} \tag{24}$$

$$\sigma'^{j+1}_n = \sigma'^j_n - \psi_{n-1/2}^{j+1}(\sigma'^{j+1}_n - \sigma'^{j+1}_{n-1})\Delta t / (\beta_n^{j+1}\Delta z^2) - \eta_n^{j+1}\Delta t / \beta_n^{j+1} \tag{25}$$

Thus, Equations (22)–(25) composed a closed system of equations, and the exact solutions for unknown variables $\sigma'^{j+1}_b$ can be obtained using the method of iteration.

In order to analyze the settlement deformation law of foundation, settlement at time $t$ is expressed as:

$$S_t = \int_0^H \varepsilon_v(z,t)\, \mathrm{d}z = \Delta z \cdot \sum_{b=1}^n \varepsilon_{v_b}^{j+1} \tag{26}$$

where $H$ is the thickness of foundation soil or the soil sample's initial height, and $\varepsilon_v(z,t)$ is the vertical strain of foundation or soil sample at any depth $z$ and time $t$.

## 5. Verification of the Suitability of UH Model

### 5.1. Verification of the Experimental Results in this Paper

First, a code is written in the Fortran software according to the above algorithm. The test results obtained for samples LB-1 and LB-2 are then used as input. Since the test sample's thickness is small, the effect of Hansbo's flow can be neglected [14] and analysis can be performed based on Darcy's flow; i.e., $m = 1.0$ or $i_1 = 0$. The compression index $C_c$, swelling index $C_s$, and secondary consolidation coefficient $C_\alpha$ are taken according to the above test results. Sorensen et al. [43] summarized some clay experimental data on clay and provided the relationship between the internal friction angle $\varphi$ and plasticity index $I_P$, as shown in Equation (27). Therefore, $\varphi = 25°$ can be estimated based on Equation (27) and Table 1 ($I_P = 17.1$), and then $M = 0.984$ can be obtained by substituting $\varphi$ into Equation (8). Moreover, the initial permeability coefficient $k_0$, permeation index $C_k$, and initial over-consolidation parameter $R_0$ were selected by trial calculation, as shown in Tables 6 and 7. Numerical simulation results are shown in Figure 9, which shows that the predicted results from the UH model, considering the time effect, are in good agreement with the experimental results. However, the predicted results of Terzaghi's theory are quite different from the experimental results in the later stage of loading. The above results fully prove the necessity of considering the soil's rheological characteristics in predicting foundation settlement in saturated clay areas, and also proved the UH model's applicability and the necessity of considering variable permeability coefficient.

$$\varphi = \begin{cases} 43 - 10 \cdot \log(I_P), & \text{(mean)} \\ 39 - 11 \cdot \log(I_P), & \text{(lower bound)} \end{cases} \tag{27}$$

**Table 6.** UH model parameters of sample LB-1.

| Loading Interval/(kPa) | $H$/mm | $e_0$ | $C_C$ | $C_k$ | $C_S$ | $C_\alpha$ | $R_0$ | $k_0$/(m/min) |
|---|---|---|---|---|---|---|---|---|
| 50–100 | 18.99 | 1.184 | | | | 0.0120 | 0.65 | $1.2 \times 10^{-8}$ |
| 100–200 | 18.01 | 1.070 | 0.42 | 0.66 | 0.100 | 0.0110 | 0.92 | $0.70 \times 10^{-8}$ |
| 200–400 | 16.63 | 0.913 | | | | 0.0091 | 0.83 | $0.40 \times 10^{-8}$ |
| 400–800 | 15.44 | 0.776 | | | | 0.0085 | 0.79 | $0.25 \times 10^{-8}$ |
| 800–1600 | 14.29 | 0.644 | | | | 0.0080 | 0.76 | $0.12 \times 10^{-8}$ |

**Table 7.** UH model parameters of sample LB-2.

| Loading Interval/(kPa) | $H$/mm | $e_0$ | $C_C$ | $C_k$ | $C_S$ | $C_\alpha$ | $R_0$ | $k_0$/(m/min) |
|---|---|---|---|---|---|---|---|---|
| 50–100 | 1.988 | 0.940 | | | | 0.0030 | 0.58 | $1.0 \times 10^{-8}$ |
| 100–200 | 1.959 | 0.911 | | | | 0.0092 | 0.53 | $0.85 \times 10^{-8}$ |
| 200–400 | 1.899 | 0.852 | 0.42 | 0.66 | 0.065 | 0.0135 | 0.65 | $0.39 \times 10^{-8}$ |
| 400–800 | 1.789 | 0.745 | | | | 0.0190 | 0.60 | $0.19 \times 10^{-8}$ |
| 800–1600 | 1.512 | 0.622 | | | | 0.0135 | 0.66 | $0.06 \times 10^{-8}$ |

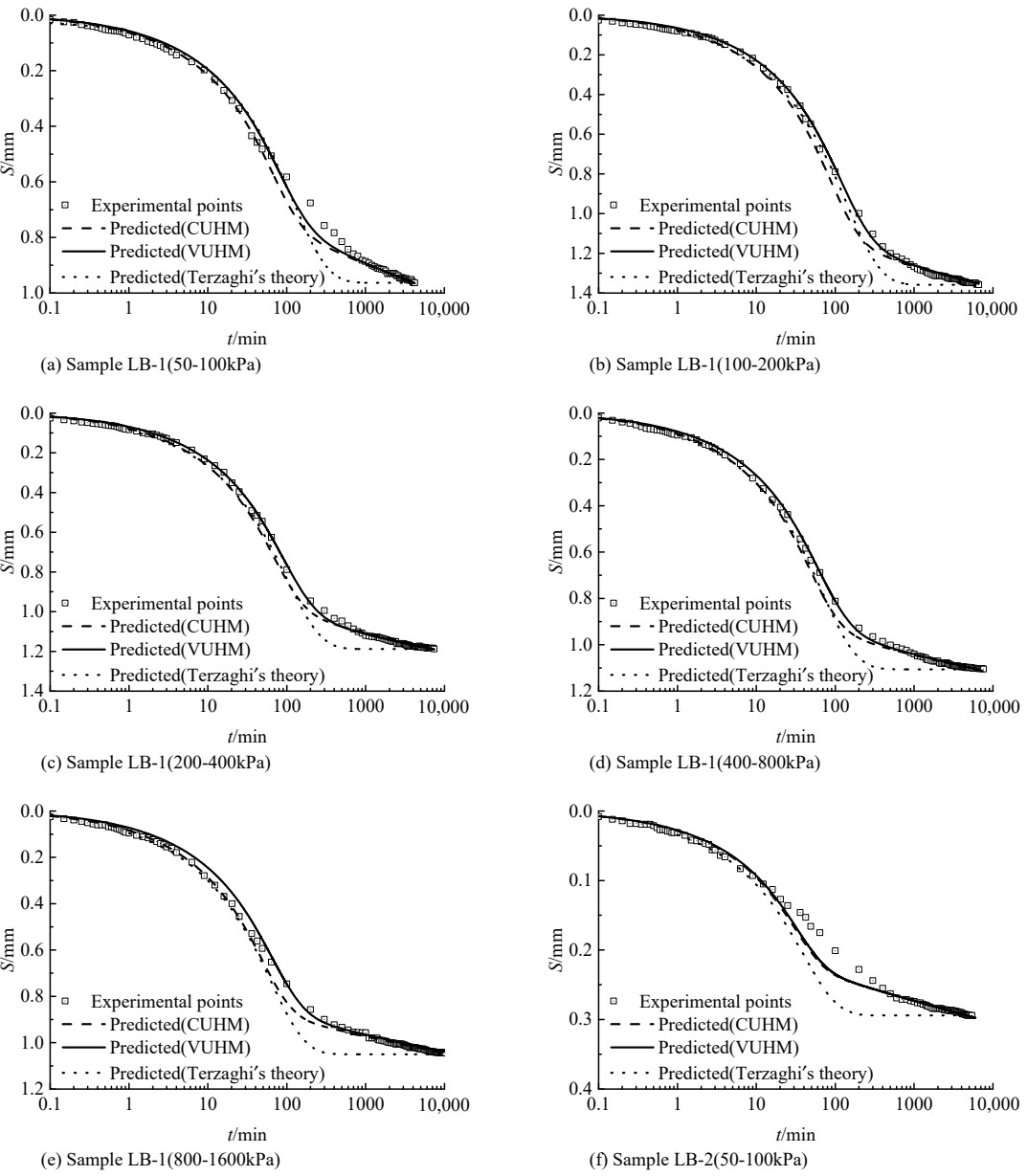

(a) Sample LB-1(50-100kPa)

(b) Sample LB-1(100-200kPa)

(c) Sample LB-1(200-400kPa)

(d) Sample LB-1(400-800kPa)

(e) Sample LB-1(800-1600kPa)

(f) Sample LB-2(50-100kPa)

**Figure 9.** *Cont.*

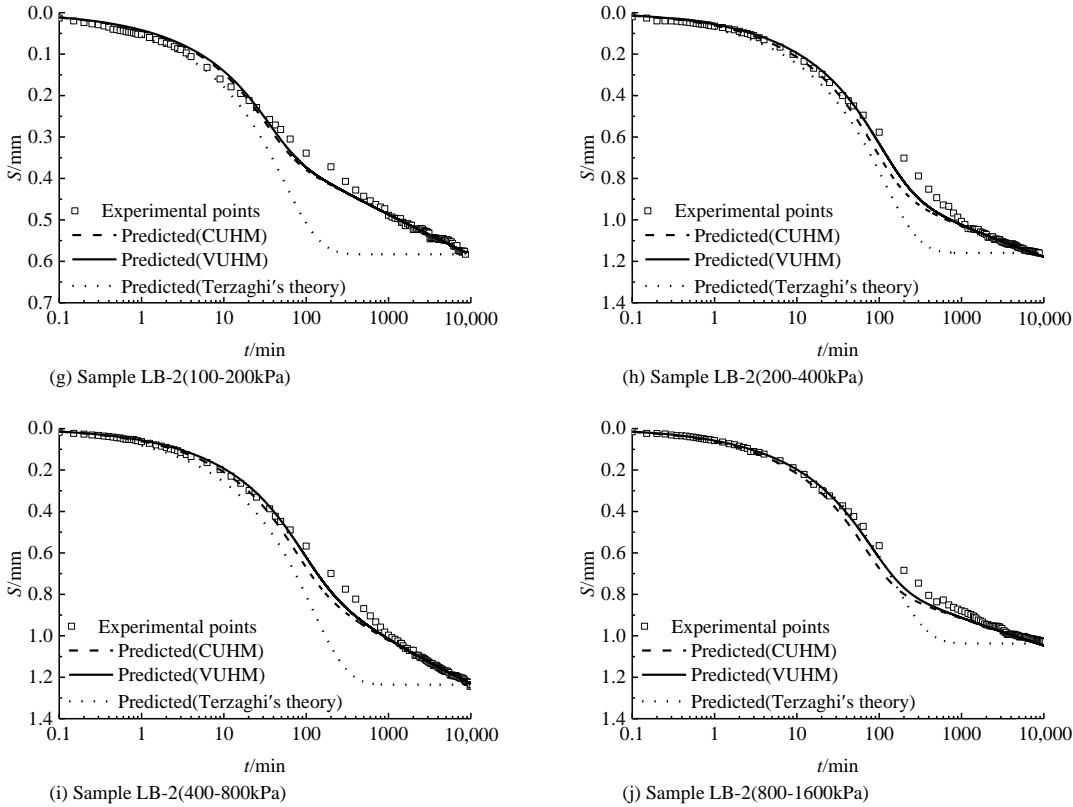

**Figure 9.** Comparison of theoretical and experimental settlement values for samples LB-1and LB-2. CUHM: unified hardening model with constant permeability coefficient, VUHM: unified hardening model with variable permeability coefficient.

## 5.2. Verification of Experimental Results obtained by Li's

Li [41] conducted one-dimensional consolidation tests on clay samples collected from Xiaoshan, Zhejiang province using an advanced GDS consolidation instrument, introduced the three-element model (TEM) and four-element model (FEM) (can been seen Figure 10) into one-dimensional consolidation analysis, and verified the obtained results by comparing them to the experimental results. Hu et al. [35] used the test data of samples 3 and 4 under loading stages of 800–1,600 kPa [41] to verify the UH model's applicability, but did not consider the permeability coefficient's influence. The same UH model parameters as those listed in [35] were used in this work as $H = 14.8$ mm, $e_0 = 0.82$, $C_C = 0.356$, $C_k = 0.528$, $C_S = 0.073$, $C_\alpha = 0.011$, $k_0 = 2.0 \times 10^{-11}$ m/s, $M = 0.984$, and $R_0 = 0.575$. The comparison of theoretical and experimental values based on the proposed model is shown with Figure 11. It can be seen that the results obtained from the UH model considering the variable permeability coefficient are closer to experimental values compared to those from the component model, which fully confirms superiority of the UH model and the necessity of considering the permeability coefficient in similar studies.

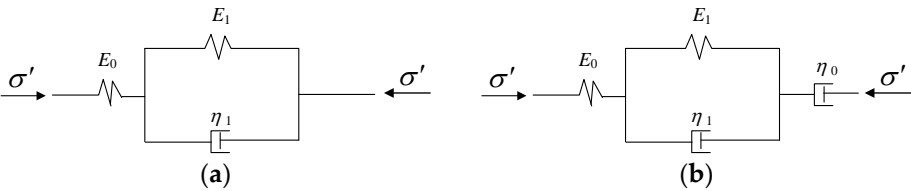

**Figure 10.** Schematic diagram of rheological model (**a**): three-element rheological model, (**b**) Four-element rheological model. where $E_0$ and $E_1$ are the moduli of the independent spring and the spring of the Kelvin–Voigt body, respectively; $\eta_0$ and $\eta_1$ are the viscosity coefficient of the independent dashpot and the dashpot of the Kelvin–Voigt body, respectively.

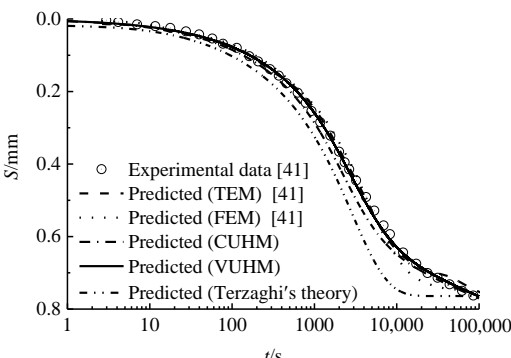

**Figure 11.** Comparison of prediction values and experimental results of the settlement theory under different models.

## 6. Conclusions

In this paper, we used an improved constant-head permeameter to measure the permeability coefficient of fine-grained soil rapidly and accurately at insignificant amounts of water flow rate. The improved oedometer is used to verify the significant rheological properties of saturated clay. Then the one-dimensional rheological consolidation equation based on Hansbo's flow and the UH model considering the time effect is derived, and the numerical solution of FVM is given. Finally, the numerical simulation results are compared to the experimental results, and the following conclusions are drawn:

(1) Compared to the traditional falling-head permeability experiment, the improved constant-head experimental device for insignificant amounts of water flow saturated clays requires shorter time and has less experimental error.

(2) The water flow of saturated clay pores deviated from Darcy flow and Hansbo's flow equations had good applicability to clay samples used in this experiment.

(3) Saturated clay had significant rheological properties. The predicted results of Terzaghi's theory were quite different from the experimental results, and the UH constitutive model agreed well with experimental results and had good applicability in predicting the rheological deformation of saturated clay samples.

The graduated U-tube with a 4 mm inner diameter and a video device were used to improve the conventional constant-head permeability test device to provide accurate and fast measurement of permeability coefficient of saturated clays under the condition of insignificant amounts of water flow, saves considerable experimental time, and avoids the experimental errors caused by environmental and human factors. The unified hardening (UH) constitutive model considering the time effect was introduced to describe the rheological properties of saturated clay soils, and Hansbo's equation was employed to describe the non-Darcian flow in the process of consolidation. Accordingly, Terzaghi's one-dimensional consolidation theory of saturated clay was modified and numerical analysis was conducted using FVM. Comparing the numerical simulation and experimental results, the applicability of the UH model considering time effects and the necessity of considering variable permeability coefficient were verified.

## 7. Patents

1. Zhongyu Liu; Yangyang Xia; Jiachao Zhang; Xinmu Zhu; Jiandong Wei. A device for preparing saturated remolded clay ring knife sample. China Patent, CN201,721,820,336.8, 2018-08-07.
2. Jiandong Wei; Yangyang Xia; Zhongyu Liu; Jiachao Zhang; Xinmu Zhu. A triple saturated clay constant-head permeability test device. China Patent, CN201,821,335,366.4, 2019-04-12.

**Author Contributions:** Conceptualization, Z.L., M.S., and Y.X.; investigation, Y.X., J.Z., and X.Z.; methodology, Z.L., M.S., Y.X., J.Z., and X.Z.; formal analysis, Z.L., M.S., and Y.X.; writing, Y.X. All the authors approved the submission of this manuscript.

**Funding:** This experiment was supported by a research grant (Project No. 51,578,511 and 51,679,219) from the National Natural Science Foundation of China (NSFC).

**Acknowledgments:** Special thanks to Yuke Wang, Jiandong Wei, and Chaojie Wang for providing instructions and suggestions on this work.

**Conflicts of Interest:** The authors declare no conflict of interest.

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
