# Peer review of "Numerical Simulation and Experiment Study on the Characteristics of Non-Darcian Flow and Rheological Consolidation of Saturated Clay"

_water, doi:10.3390/w11071385_

Round 1

Reviewer 1 Report

The paper deals with the one-dimensional consolidation of saturated clays. In particular, the paper is aimed to show the importance of considering a non-Darcian flow for pore water and the rheological properties of the soil. Constant-head permeability tests and oedometer tests were carried out to this end. In addition, Hansbo's equation and a unified hardening (UH) constitutive model are used to simulate the experimental data derived from clay samples. A comparison is presented between experimental data and theoretical results obtained using the proposed approach. The work is interesting. However, there are several points that should be adequately addressed before considering the paper for publication. In particular, the authors should clarify the following points:

1) On 3rd line from the bottom of page 1, the preposition “for” should be deleted.

2) On page 2, what is “cohesion of pore water”? Please clarify.

3) Considering that other factors could significantly affect soil consolidation, such as anisotropy and non-linearity of the soil, the authors should include in the “Introduction” a sentence concerning these aspects. In this connection, reference could be made to the following papers:

Davis, E. H., and Raymond, G. P. (1965). A non-linear theory of consolidation. GĂ©otechnique, 15, 161-173.

Conte E. (1998). Consolidation of anisotropic soil deposits. Soils and Foundations, 38(4), 227-237.

Bartholomeeusen G., Sills G. C., Znidarčiċ D., Van Kesteren W., Merckelbach L. M., Pyke R., Carrier III W. D., Lin H., Penumadu D., Winterwerf H., Masala S. and Chan D. (2002). Sidere: numerical prediction of large-strain consolidation. Géotechnique, 52(9), 639-648.

Zhuang, Y. C., Xie, K. H., and Li, X. B. (2005). Nonlinear analysis of consolidation with variable compressibility and permeability. Journal of Zhejiang University (Engineering Science), 6A, 181-187.

Conte E., Troncone A. (2007). Nonlinear consolidation of thin layers subjected to time-dependent loading. Canadian Geotechnical Journal, 44, 717-725.

4) The authors have used an improved constant-head test device to measure the hydraulic conductivity of the soil samples (Fig. 1).  However, a constant-head permeability test is generally used for high permeable soils. On the contrary, the soils tested in the paper are clayey soils with very low values of the hydraulic conductivity (Table 1). Therefore, the testing apparatus used by the authors should not be in principle suitable for testing the soils considered in the paper. It is not clear how the improvements proposed by the authors can overcome this limitation. A comparison with some results from falling-head permeability tests (if available) could be useful to clarify this point.

5) The authors assert that the tests were carried out on saturated remoulded clays. However, remoulding effects could significantly affected soil properties and reliability of the measured parameters. Can the conclusions of the study be extended to a “undisturbed” soil as well? Again, a comparison with experimental results from undisturbed soils (if available) could be useful to clarify this point.

6) How Hansbo's flow parameters in Table 2 were obtained? Was a trial and error procedure used to this end?

7) The values of the hydraulic gradient in Table 2 and in Fig. 2 are very high if compared with those usually found in the real situations. As shown by Eq. (1) and the results of Fig. 2, for low values of the gradient, Darcy’s law provides an adequate description of the water flow from a practical viewpoint (also the authors have used Darcy’s law in their numerical simulations). In conclusion, what is the necessity of using Eq. (1) instead of Darcy's law?

8) The authors should also provide a more detailed description of the improved consolidation apparatus used in the paper.

9)  Eq (11) seems be incorrect. Since an external load is instantaneously applied (and then held constant with time) under one-dimensional conditions, po should be deleted in this equation (in fact,  po=u).

10) What is q in Eq. (13)?

11) How the constitutive parameter M was evaluated? Was this parameter obtained exprimentally or was assumed? Please, complete.

12) A comparison between experimental results and those provided by the classical Terzaghi solution could be useful to show the effects of the rheological properties of the soil and corroborate the validity of the proposed approach.

In the light of these comments, in my opinion, the paper cannot be considered for publication in the present form. 

Author Response

Dear Editors and Reviewers:

Thank you for your letter and for the reviewers’ comments concerning our manuscript entitled “Investigation on the Characteristics of Non-Darcian Flow and Rheological Consolidation of Saturated Clay” (ID: water-519000). Those comments are all valuable and very helpful for revising and improving our paper, as well as the important guiding significance to our researches. We have studied comments carefully and have made correction which we hope meet with approval. Revised portion are marked in red in the paper. The main corrections in the paper and the responds to there viewer’s comments are as flowing: Responds to the reviewer’s comments:

Reviewer #1:

Point 1: On 3rd line from the bottom of page 1, the preposition “for” should be deleted.

Response: Thank you for your valuable comment, we are very sorry for our incorrect writing, on 3rd line from the bottom of page 1, the preposition “for” has been deleted.

Point 2: On page 2, what is “cohesion of pore water”? Please clarify.

Response: Thank you for your valuable comment, we are very sorry for our incorrect writing, on page 2, "cohesion of pore water" is replaced by "viscosity of water around soil particles ". Here, "viscosity " is mainly caused by the bound water.

Point 3: Considering that other factors could significantly affect soil consolidation, such as anisotropy and non-linearity of the soil, the authors should include in the “Introduction” a sentence concerning these aspects. In this connection, reference could be made to the following papers: 

Davis, E. H., and Raymond, G. P. (1965). A non-linear theory of consolidation. GĂ©otechnique, 15, 161-173.

Conte E. (1998). Consolidation of anisotropic soil deposits. Soils and Foundations, 38(4), 227-237.

Bartholomeeusen G., Sills G. C., Znidarčiċ D., Van Kesteren W., Merckelbach L. M., Pyke R., Carrier III W. D., Lin H., Penumadu D., Winterwerf H., Masala S. and Chan D. (2002). Sidere: numerical prediction of large-strain consolidation. Géotechnique, 52(9), 639-648.

Zhuang, Y. C., Xie, K. H., and Li, X. B. (2005). Nonlinear analysis of consolidation with variable compressibility and permeability. Journal of Zhejiang University (Engineering Science), 6A, 181-187.

Conte E., Troncone A. (2007). Nonlinear consolidation of thin layers subjected to time-dependent loading. Canadian Geotechnical Journal, 44, 717-725.

Response: Thank you for your valuable comment, we are very sorry for our negligence of anisotropy and non-linearity of the soil, on page 2, the author has summarized these aspects and marked them in red.

Point 4: The authors have used an improved constant-head test device to measure the hydraulic conductivity of the soil samples (Fig. 1). However, a constant-head permeability test is generally used for high permeable soils. On the contrary, the soils tested in the paper are clayey soils with very low values of the hydraulic conductivity (Table 1). Therefore, the testing apparatus used by the authors should not be in principle suitable for testing the soils considered in the paper. It is not clear how the improvements proposed by the authors can overcome this limitation. A comparison with some results from falling-head permeability tests (if available) could be useful to clarify this point.

Response: Thank you for your valuable comment, we have made correction according to the Reviewer’s comment. On page 3, the falling-head available test results are added in the paper as shown in Figure 1. , it can be seen that the test takes a long time and is easily influenced by human or environmental factors,. Using a graduated U-tube with 4 mm inner diameter and a video device are used to improve the constant-head permeability tests device, which realizes the accurate and fast measurement of permeability coefficient of saturated clay under the condition of insignificant amounts of water flow, saves a lot of time for the experiment and avoids the experimental errors caused by environmental and human factors. Furthermore, on page4, we re-written the detailed usage method of the improved constant-head permeability tests device. The detailed test procedures are as follows: (1) Firstly, a layer of Vaseline is coated on the inner wall of the permeameter sleeve, and a filter paper is attached to the upper and lower surfaces of the saturated sample with cutting ring (to avoid being covered by Vaseline when the sample was installed, which affects the permeability of the sample); then Loaded the sample into the permeameter sleeve, wiped out the excess Vaseline and replaced the upper and lower filter paper; Finally, install the permeameter sleeve on the base and tighten the screw. (2) Adjust the constant head control device and discharge excess air inside the permeameter, then adjust the liquid level to any calibration line in graduated U-tube. (3) Open the left valve on the bottom of the permeameter and the video device on the right side, wait for the liquid level in the U-tube to stop the test on the next calibration line ( where one scale is 0.034 ml, therefore the test time is shorter ); then adjust the constant head control device, using the same method the corresponding seepage velocities under different constant pressure heads were obtained, and then v-i curve were obtained by connecting these experimental points.

Point 5: The authors assert that the tests were carried out on saturated remoulded clays. However, remoulding effects could significantly affected soil properties and reliability of the measured parameters. Can the conclusions of the study be extended to a “undisturbed” soil as well? Again, a comparison with experimental results from undisturbed soils (if available) could be useful to clarify this point.

Response: Thank you for your valuable comment, as reviewer suggested that this paper only studies saturated remolded clay, but there may be differences in properties and parameters between remolded clay and undisturbed clay, which will be the next research direction of this paper.

Point 6: How Hansbo's flow parameters in Table 2 were obtained? Was a trial and error procedure used to this end?

Response: Thank you for your valuable comment, we are very sorry for our negligence the method of obtaining Hansbo's flow parameters. On page 4, we re-written the detailed fitting method. The specific fitting method is as follows: (1) If there are n experimental points, firstly use the Origin software to non-linearly fitted 0 to g (where 0<g<n) experimental points using formula  to obtain the values of c and m. (2) The formula  was obtained by substituting c and m into formula , then the g to n experimental points were non-linearly fitted through formula  by Origin software, and the value of i1 was obtained. (3) By substituting the values of c, m and i1 into the Eq. (1), the calculation points of Hansbo's flow mathematical model are obtained, and then the correlation between the experimental values and the calculation values was compared, the maximum R2 is required. Otherwise, the above methods are repeated at 0 to g+1 and g+1 to n experimental points.

Point 7: The values of the hydraulic gradient in Table 2 and in Fig. 2 are very high if compared with those usually found in the real situations. As shown by Eq. (1) and the results of Fig. 2, for low values of the gradient, Darcy’s law provides an adequate description of the water flow from a practical viewpoint (also the authors have used Darcy’s law in their numerical simulations). In conclusion, what is the necessity of using Eq. (1) instead of Darcy's law?

Response: Thank you for your valuable comment. Due to the lack of relevant experimental data, there is no unified statement on the range of i0 in the curren studies, Hansbo[3] stated that i0 ranged from 1 to 4, Dubin et al. [8] reported that was between 3 to 25, Sun et al. [11] reported that was between 0.5 to 14. On page 13, the purpose of this paper is only to validate the applicability of UH model considering time effect by numerical simulation. Literature[14]Xie.et al. has emphasized the necessity of non-Darcian flow instead of Darcy's flow, and the increase of Hansbo¢s flow parameters m and i1 will delay the consolidation rate of saturated clay. However, when the thickness of soil layer is very thin, the influence of non-Darcian flow can be ignored. In this paper, the height of the test sample is 2 cm, so it can be calculated by Darcy’s law.

References:

3. Hansbo S. Consolidation of clay with special reference to influence of vertical sand drain. Swedish Geotechnical Institute, 1960, 18, 45-50.

1. Dubin B.; Moulin G. Influence of a critical gradient on the consolidation of clays. Consolidation of soils: testing and evaluation(STP 892). West Conshochen(PA): ASTM, 1986, 354-377.

11. Sun L.Y.; Yue J.C.; Zhang J. Experimental study on non-Darcian permeability characteristics of saturated clays. Journal of Zhengzhou University (Engineering Science), 2010, 31, 31-34.

14. Li C.X.; Xie K.H. One-dimensional nonlinear consolidation of soft clay with the non-Darcian flow. Journal of Zhejiang University (Science A), 2013, 14, 435-446.

Point 8: The authors should also provide a more detailed description of the improved consolidation apparatus used in the paper.

Response: Thank you for your valuable comment, we are very sorry for our negligence the detailed method of the improved consolidation apparatus, on page 6, we re-written the detailed method of the improved consolidation apparatus. Detailed test procedures are as follows: (1) Firstly, the prepared saturated clay samples LB-1 and LB-2 were loaded into the consolidation cylinder, respectively; as shown in Figure 5., filter paper and permeable plate were placed on the top and bottom of the samples. (2) Running data collection software on computer to adjust the displacement meter and pore pressure sensor, then start loading, when the displacement sensor reads less than 0.1 mm in a day or the pore pressure dissipates more than 95%, the next stage of loading or unloading was carried out according to Table 4. (3) According to displacement sensor reading, the vertical settlement deformation S of soil sample was calculated, and then the corresponding void ratio e can be obtained according to Eq. (3). 

                                  (3)

where e0 is the initial void ratio corresponding to the initial thickness H0 of the soil sample.

Point 9: Eq (11) seems be incorrect. Since an external load is instantaneously applied (and then held constant with time) under one-dimensional conditions, p0 should be deleted in this equation (in fact,  p0=u).

Response: Thank you for your valuable comment, Here, the p0 is the external load, not the initial load. In this paper, the instantaneous load is still used, p0 is the ultimate load applied on the soil layer.

Point 10: What is q in Eq. (13)?

Response: Thank you for your valuable comment, we are very sorry for our negligence introduction to q , on page 10, we added an introduction to q. where q is the change of water quantity within the unit cell during an infinitesimal time period of dt can be expressed as , A is the area of water flowing through the unit cell.

Point 11: How the constitutive parameter M was evaluated? Was this parameter obtained exprimentally or was assumed? Please, complete.

Response: Thank you for your valuable comment, Sorensen et al. [43]summarized some clay experimental data provided the relationship between internal friction angle j and plasticity index IP., as shown in Eq. (28), so j = 25° can be estimated based on Eq. (28) and Table 1. (IP=17.1) , and then M = 0.984 can be obtained by substituting j into Eq. (8).

                          (27)

           (8)

Table 1.  Physical properties of soils

Specific gravity, Gs

Liquid limit,

wL (%)

Plastic limit,

wP (%)

Plastic index,

IP (%)

2.71

39.4

22.3

17.1

References:

43. Sorensen, K. K.; Okkels, N. Correlation between Drained Shear Strength and Plasticity Index of Undisturbed Overconsolidated Clays, Proceedings, 18th International Conference on Soil Mechanics and Geotechnical Engineering, Paris, Presses des Ponts, 2013, 1, 423-428. 

Point 12: A comparison between experimental results and those provided by the classical Terzaghi solution could be useful to show the effects of the rheological properties of the soil and corroborate the validity of the proposed approach.

Response: Thank you for your valuable comment, we have made correction according to the Reviewer’s comment, on page14-15, as shown in Figures 9. and 10. we provided the solution of classical Terzaghi's theory.

We tried our best to improve the manuscript and madesome changes in the manuscript. These changes will not influence thecontent and framework of the paper. And here we did not list the changes butmarked in red in revised paper. We appreciate for Editors/Reviewers’ warm workearnestly, and hope that the correction will meet with approval. In addition, we are willing to pay extra for MDPI to provide English editing services. Once again, thank you very much for your comments and suggestions.

Looking forward to hearing from you. Thank you and best regards.

 Yours sincerely, 

Zhongyu Liu

June 18, 2019

School of Civil Engineering

Zhengzhou University

Reviewer 2 Report

This paper explores the permeability and rheological consolidation characteristics of saturated clay by using the improved 1D consolidation test devices and the UH model considering time effect. Basically, this paper is short of demonstrating its novelty and significance. Here are the detailed comments.

1.       In section 2.1, it's not clear why constant head method is not suitable for saturated clay and how to improve it. (1) video recorder and measurement method seem also suitable for sand and silt (2) the uniqueness and specialty of using these two improvements need more details

2.       The title “Investigation on the Characteristics of Non-Darcian Flow and Rheological Consolidation of Saturated Clay” suggests more analyzes and interpretations on the characteristics using new theory or new hypothesis or new observations, but in the paper the first part basically described the improved devices and the test results obtained by these devices; and the second part introduced a 1D model based on UH model. These two parts are basically describing the “tools” for investigations. The title is suggested to be changed.

3.       The novelty and significance of this study haven’t been demonstrated well and needs more emphasizes. (1) the devices was improved. Please show the readers why? (2) the tests results obtained from the improved devices may be compared with the results obtained from the traditional devices using Figures and Tables for clear demonstrations. (3) the advantages of the improved devices can be discussed in more details.

4.       The discretization method for Eq. (14-17) may need more details. (1) is it Finite Difference method or Finite volume method? (2) are Eq. (19)-(22) solved explicitly or implicitly?

5.       The conclusions are weak and needs to be re-written.

6.       Others:

(1)  On page 1, “..different mathematical equations for were…”, please delete “for”.

(2)  On page 10, “discrete” may be changed into “discretized”.

On page 12, please add abbreviations for three-element model and four-element model as references for Fig. 9.  “three-element model (TEM) and four-element model (FEM)”

Author Response

Dear Editors and Reviewers:

Thank you for your letter and for the reviewers’ comments concerning our manuscript entitled “Investigation on the Characteristics of Non-Darcian Flow and Rheological Consolidation of Saturated Clay” (ID: water-519000). Those comments are all valuable and very helpful for revising and improving our paper, as well as the important guiding significance to our researches. We have studied comments carefully and have made correction which we hope meet with approval. Revised portion are marked in red in the paper. The main corrections in the paper and the responds to there viewer’s comments are as flowing: Responds to the reviewer’s comments:

Reviewer #2:

Point 1: In section 2.1, it's not clear why constant head method is not suitable for saturated clay and how to improve it. (1) video recorder and measurement method seem also suitable for sand and silt (2) the uniqueness and specialty of using these two improvements need more details

Response: Thank you for your valuable comment, we have made correction according to the Reviewer’s comment. On page4, we re-written the detailed usage method of the improved constant-head permeability tests device. Detailed test procedures are as follows: (1) Firstly, a layer of Vaseline was coated on the inner wall of the permeameter sleeve, and a filter paper was attached to the upper and lower surfaces of the saturated sample with ring knife (to avoid being covered by Vaseline when the sample was installed, which affects the permeability of the sample); then the sample was loaded into the permeameter sleeve, the excess Vaseline was wiped out and the upper and lower filter paper was replace; finally, install the permeameter sleeve on the base and tighten the screw. (2) Adjust the constant head control device and discharge excess air inside the permeameter, then adjust the liquid level to any calibration line in graduated U-tube. (3) Open the left valve on the bottom of the permeameter and the video device on the right side, wait for the liquid level in the U-tube to stop the test on the next calibration line ( where one scale is 0.034 ml, therefore the test time is shorter ); then adjust the constant head control device, using the same method the corresponding seepage velocities under different constant pressure heads were obtained, and then v-i curve were obtained by connecting these experimental points.

Point 2: The title “Investigation on the Characteristics of Non-Darcian Flow and Rheological Consolidation of Saturated Clay” suggests more analyzes and interpretations on the characteristics using new theory or new hypothesis or new observations, but in the paper the first part basically described the improved devices and the test results obtained by these devices; and the second part introduced a 1D model based on UH model. These two parts are basically describing the “tools” for investigations. The title is suggested to be changed.

Response: Thank you for your valuable comment, we have made correction according to the Reviewer’s comment, we changed the title to” Numerical Simulation and Experiment Study on the Characteristics of Non-Darcian Flow and Rheological Consolidation of Saturated Clay”, and divided section 3 into sections “3. Experimental study on rheological consolidation characteristics of saturated clay”and “4. Derivation of governing equations” and “5.Verification of the suitability of UH model ”.

Point 3: The novelty and significance of this study haven’t been demonstrated well and needs more emphasizes. (1) the devices was improved. Please show the readers why? (2) the tests results obtained from the improved devices may be compared with the results obtained from the traditional devices using Figures and Tables for clear demonstrations. (3) the advantages of the improved devices can be discussed in more details.

Response: Thank you for your valuable comment, we have made correction according to the Reviewer’s comment. 

(1) The devices was improved. Please show the readers why? And (2) the tests results obtained from the improved devices may be compared with the results obtained from the traditional devices using Figures and Tables for clear demonstrations.

Response: On page 3, the falling-head test device and available test results shown in Figure 1. are added in the paper, it can be seen that the test takes a long time and is easily influenced by human or environmental factors.

(3) The advantages of the improved devices can be discussed in more details.

Response: Using a graduated U-tube with 4 mm inner diameter and a video device are used to improve the constant-head permeability tests device, which realizes the accurate and fast measurement of permeability coefficient of saturated clay under the condition of insignificant amounts of water flow, saves a lot of time for the experiment and avoids the experimental errors caused by environmental and human factors.

Point 4: The discretization method for Eq. (14-17) may need more details. (1) is it Finite Difference method or Finite volume method? (2) are Eq. (19)-(22) solved explicitly or implicitly?

Response: Thank you for your valuable comment, we are very sorry for our incorrect writing, on pages 11 and 12, we add “4.3 Discretization of the governing equation” to introduce the process of solving finite volume method(FVM) in detail.

Point 5: The conclusions are weak and needs to be re-written.

Response: Thank you for your valuable comment, we have made correction according to the Reviewer’s comment, on page 16, we re-written the conclusion of this paper, at the end of the conclusion, the idea of this paper is given.

Point 6: (Others:

(1) On page 1, “..different mathematical equations for were…”, please delete “for”.

(2) On page 10, “discrete” may be changed into “discretized”.

(3)On page 12, please add abbreviations for three-element model and four-element model as references for Fig. 9.  â€śthree-element model (TEM) and four-element model (FEM)”)

Response: Thank you for your valuable comment, we are very sorry for our incorrect writing,(1) on page 1, the preposition “for” has been deleted. (2) On page 10, “discrete” has been changed into “discretized”.(3) On page 15, we have added abbreviations for three-element model and four-element model.

We tried our best to improve the manuscript and madesome changes in the manuscript. These changes will not influence thecontent and framework of the paper. And here we did not list the changes butmarked in red in revised paper. We appreciate for Editors/Reviewers’ warm workearnestly, and hope that the correction will meet with approval. In addition, we are willing to pay extra for MDPI to provide English editing services. Once again, thank you very much for your comments and suggestions.

Looking forward to hearing from you. Thank you and best regards.

 Yours sincerely, 

Zhongyu Liu

June 18, 2019

School of Civil Engineering

Zhengzhou University

Round 2

Reviewer 1 Report

The authors have made the changes required by me. Therefore, in my opinion, the paper could be accepted for publication.

Author Response

Dear Editors and Reviewers:

Thank you for your letter and for the reviewers’ comments concerning our manuscript entitled “Numerical Simulation and Experiment Study on the Characteristics of Non-Darcian Flow and Rheological Consolidation of Saturated Clay” (ID: water-519000). Those comments are all valuable and very helpful for revising and improving our paper, as well as the important guiding significance to our researches. We have studied comments carefully and have made correction which we hope meet with approval. Revised portion are marked in red in the paper. The main corrections in the paper and the responds to there viewer’s comments are as flowing: Responds to the reviewer’s comments:

Reviewer #1:

Open Review (x) I would not like to sign my review report

( ) I would like to sign my review report

English language and style ( ) Extensive editing of English language and style required

( ) Moderate English changes required

(x) English language and style are fine/minor spell check required

( ) I don't feel qualified to judge about the English language and style

Yes Can be improved Must be improved Not applicable

Does the introduction provide sufficient background and include all relevant references? (x) ( ) ( ) ( )

Is the research design appropriate? (x) ( ) ( ) ( )

Are the methods adequately described? (x) ( ) ( ) ( )

Are the results clearly presented? (x) ( ) ( ) ( )

Are the conclusions supported by the results? (x) ( ) ( ) ( )

Comments and Suggestions for Authors

The authors have made the changes required by me. Therefore, in my opinion, the paper could be accepted for publication.

We tried our best to improve the manuscript and madesome changes in the manuscript. These changes will not influence thecontent and framework of the paper. And here we did not list the changes butmarked in red in revised paper. We appreciate for Editors/Reviewers’ warm workearnestly, and hope that the correction will meet with approval. In addition, we are willing to pay extra for MDPI to provide English editing services. Once again, thank you very much for your comments and suggestions.

Looking forward to hearing from you. Thank you and best regards.

 Yours sincerely, 

Yangyang Xia

June 27, 2019

School of Civil Engineering

Zhengzhou University

Reviewer 2 Report

This revision has been improved significantly compared to the original version. Basically I recommend it for publication. There are some minor issues:

(1) On page 4, Eq. (1) may be introduced first and then the fitting method.

(2) On page 12, "Eq. (21) can be sorted out as" should be "Eq. (22) can be...."

(3) On page 13, the sentence "Terzagh's theory are.....Where fully proves..." is not understandable. There must be typos. "Where" should be "which"?? Please clarify it.

(4) On page 15, right after "three-elements and four-element models" please add "TEM" and "FEM" as references.

Author Response

Dear Editors and Reviewers:

Thank you for your letter and for the reviewers’ comments concerning our manuscript entitled “Numerical Simulation and Experiment Study on the Characteristics of Non-Darcian Flow and Rheological Consolidation of Saturated Clay” (ID: water-519000). Those comments are all valuable and very helpful for revising and improving our paper, as well as the important guiding significance to our researches. We have studied comments carefully and have made correction which we hope meet with approval. Revised portion are marked in red in the paper. The main corrections in the paper and the responds to there viewer’s comments are as flowing: Responds to the reviewer’s comments:

Reviewer #2:

Open Review (x) I would not like to sign my review report

( ) I would like to sign my review report

English language and style ( ) Extensive editing of English language and style required

( ) Moderate English changes required

(x) English language and style are fine/minor spell check required

( ) I don't feel qualified to judge about the English language and style

Yes Can be improved Must be improved Not applicable

Does the introduction provide sufficient background and include all relevant references? (x) ( ) ( ) ( )

Is the research design appropriate? (x) ( ) ( ) ( )

Are the methods adequately described? (x) ( ) ( ) ( )

Are the results clearly presented? (x) ( ) ( ) ( )

Are the conclusions supported by the results? (x) ( ) ( ) ( )

Comments and Suggestions for Authors

This revision has been improved significantly compared to the original version. Basically I recommend it for publication. There are some minor issues:

Point 1: On page 4, Eq. (1) may be introduced first and then the fitting method.

Response: Thank you for your valuable comment. On page4, we have made correction according to the Reviewer’s comment.

Point 2: On page 12, "Eq. (21) can be sorted out as" should be "Eq. (22) can be...."

Response: Thank you for your valuable comment. On page12, we have made correction according to the Reviewer’s comment.

Point 3: On page 13, the sentence "Terzagh's theory are.....Where fully proves..." is not understandable. There must be typos. "Where" should be "which"?? Please clarify it.

Response: Thank you for your valuable comment, we are very sorry for our incorrect writing. On page 13, we have re-edited the remark mentioned by the expert reviewer. If it is improper, please correct it.

Point 4: On page 15, right after "three-elements and four-element models" please add "TEM" and "FEM" as references.

Response: Thank you for your valuable comment, we are very sorry for our incorrect writing. On page 15, we have added "TEM" and "FEM" after "three-elements and four-element models". If it is improper, please correct it.

We tried our best to improve the manuscript and madesome changes in the manuscript. These changes will not influence thecontent and framework of the paper. And here we did not list the changes butmarked in red in revised paper. We appreciate for Editors/Reviewers’ warm workearnestly, and hope that the correction will meet with approval. In addition, we are willing to pay extra for MDPI to provide English editing services. Once again, thank you very much for your comments and suggestions.

Looking forward to hearing from you. Thank you and best regards.

 Yours sincerely, 

Yangyang Xia

June 27, 2019

School of Civil Engineering

Zhengzhou University
